# Benchmarking Generative Latent Variable Models for Speech

**Jakob D. Havtorn**[1,3]**, Lasse Borgholt**[2,3]**, Jes Frellsen**[1]**, Søren Hauberg**[1]**, Lars Maaløe**[1,3]

[1]Department of Applied Mathematics and Computing, Technical University of Denmark
[2]Department of Computer Science, University of Copenhagen, Denmark
[3]Corti, Copenhagen, Denmark
`jdh@corti.ai`

## Abstract

Stochastic latent variable models (LVMs) achieve state-of-the-art performance on natural image generation but are still inferior to deterministic models on speech. In this paper, we develop a speech benchmark of popular temporal LVMs and compare them against state-of-the-art deterministic models. We report the likelihood, which is a much used metric in the image domain, but rarely, or incomparably, reported for speech models. To assess the quality of the learned representations, we also compare their usefulness for phoneme recognition. Finally, we adapt the Clockwork VAE, a state-of-the-art temporal LVM for video generation, to the speech domain. Despite being autoregressive only in latent space, we find that the Clockwork VAE can outperform previous LVMs and reduce the gap to deterministic models by using a hierarchy of latent variables.

## 1 Introduction

Since their introduction, temporal latent variable models (LVMs) for speech (Chung et al., 2015; Fraccaro et al., 2016) based on the variational autoencoder (VAE, Kingma & Welling, 2014; Rezende et al., 2014) have seen little development compared to their image domain counterparts. While LVMs now achieve superior likelihoods on images compared to deterministic models (Child, 2021; Sinha & Dieng, 2021; Kingma et al., 2021), they are still inferior to deterministic models such as WaveNet on speech (van den Oord et al., 2016). Development in the image domain has been driven by established likelihood benchmarks, but in the speech domain, likelihoods are often not reported (van den Oord et al., 2016; Hsu et al., 2017; van den Oord et al., 2018b) or are incomparable due to subtle differences in the chosen data distributions (Chung et al., 2015; Fraccaro et al., 2016; Hsu et al., 2017; Aksan & Hilliges, 2019). This makes it hard to develop explicit likelihood models for speech.

In this paper, we develop a likelihood benchmark for recent temporal LVMs and compare to deterministic counterparts including WaveNet. We introduce a hierarchical LVM without autoregressive decoder and finally evaluate the learned representations for phoneme recognition. We find that (i) LVMs achieve likelihoods superior to WaveNet at low temporal resolution, (ii) LVMs with autoregressive decoders achieve better likelihoods than a non-autoregressive LVM, (iii) LVM likelihoods improve when using hierarchies of latent variables as also seen for images and, (iv) LVM representations are as good or better than Mel spectrograms for phoneme recognition.

**Scope and related work.** At a high level, this benchmark brings order to LVM model comparisons for speech and also provides useful reference implementations of the models[1]. An extended version of this paper is available at arXiv:2202.12707. LVMs based on the VAE are of interest due to their ability to learn an approximate posterior distribution over latent variables. This makes them useful beyond generation for e.g. semi-supervised learning (Kingma et al., 2014) and anomaly detection (Havtorn et al., 2021). In this paper, we focus on the VRNN (Chung et al., 2015), SRNN (Fraccaro et al., 2016) and STCN (Aksan & Hilliges, 2019) since they are similar to the original VAE framework. We exclude some related work from the benchmark. Specifically, the FH-VAE (Hsu et al., 2017) adds a global latent variable but we exclude it due to it's segmentation of the training data. We also exclude

---

[1] `github.com/JakobHavtorn/benchmarking-lvms`

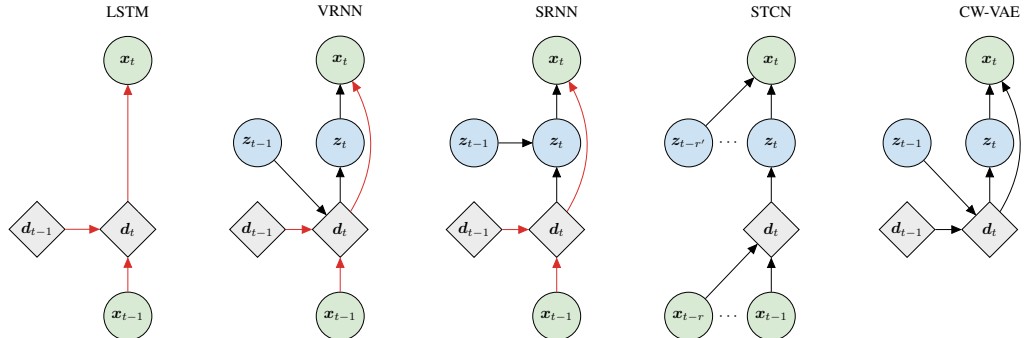

Figure 1: Generative models of an LSTM, VRNN, SRNN, STCN and CW-VAE for a single time step. The STCN and CW-VAE are illustrated with a single latent variable. Red arrows indicate purely deterministic paths from the output $\boldsymbol{x}_t$ to previous input $\boldsymbol{x}_{<t}$ without passing a stochastic node. We provide additional graphical illustrations including inference models in appendix J.

Z-forcing (Goyal et al., 2017) which resembles the SRNN, but deviates from maximum likelihood training by using an auxiliary loss. We exclude the VQ-VAE (van den Oord et al., 2018b) which is a hybrid LVM and autoregressive model, since it uses a quantized latent space. Finally, we exclude the Stochastic WaveNet (Lai et al., 2018) which is similar, but inferior, to the STCN (Aksan & Hilliges, 2019).

All selected models have autoregressive generative models. We therefore formulate and benchmark a novel temporal LVM which does not use an autoregressive decoder. We do so by adapting the hierarchical Clockwork Variational Autoencoder (Saxena et al., 2021), originally proposed for video generation, to speech. Before presenting the results, we provide a brief survey of existing LVMs for speech in a coherent notation.

## 2 MODELS

**Sequential deep latent variable models.** The models we consider are all sequential deep latent variable models trained with variational inference and the reparameterization trick (Kingma & Welling, 2014). The input is a variable-length sequence $\boldsymbol{x} = \boldsymbol{x}_{1:T} = (\boldsymbol{x}_1, \boldsymbol{x}_2, \ldots, \boldsymbol{x}_T)$ with $\boldsymbol{x}_t \in \mathbb{R}^{D_x}$. The models first encode $\boldsymbol{x}_{1:T}$ to a temporal posterior distribution $q(\boldsymbol{z}|\boldsymbol{x})$ and sample a latent representation $\boldsymbol{z}_{1:T}$ where $\boldsymbol{z}_t \in \mathbb{R}^{D_z}$. This is then used to reconstruct the input while the posterior distribution is regularized to be close to a prior distribution $p(\boldsymbol{z}_t|\cdot)$ where the dot indicates that it may depend on latent and observed variables at previous time steps, $\boldsymbol{z}_{<t}$ and $\boldsymbol{x}_{<t}$ where $\boldsymbol{z}_{<t} := (\boldsymbol{z}_1, \boldsymbol{z}_2, \ldots, \boldsymbol{z}_{t-1})$. The models are trained to maximize a lower bound $\mathcal{L}(\boldsymbol{\theta}, \boldsymbol{\phi}; \boldsymbol{x})$ on the marginal likelihood $\log p_{\boldsymbol{\theta}}(\boldsymbol{x})$,

$$\log p_{\boldsymbol{\theta}}(\boldsymbol{x}) = \log \int p_{\boldsymbol{\theta}}(\boldsymbol{x}, \boldsymbol{z}) \, \mathrm{d}\boldsymbol{z} \geq \int q_{\boldsymbol{\phi}}(\boldsymbol{z}|\boldsymbol{x}) \log \frac{p_{\boldsymbol{\theta}}(\boldsymbol{x}|\boldsymbol{z})p(\boldsymbol{z})}{q_{\boldsymbol{\phi}}(\boldsymbol{z}|\boldsymbol{x})} \, \mathrm{d}\boldsymbol{z} \triangleq \mathcal{L}(\boldsymbol{\theta}, \boldsymbol{\phi}; \boldsymbol{x}) \ , \quad (1)$$

where $\boldsymbol{\theta}$ are parameters of the generative model and $q_{\boldsymbol{\phi}}(\boldsymbol{z}|\boldsymbol{x})$ is the variational approximation to the true posterior. Graphical illustrations of the models can be seen in figure 1 and appendix J.

**Variational recurrent neural network (VRNN).** The VRNN (Chung et al., 2015) is essentially a VAE per timestep. As seen in figure 1, it is conditioned on the hidden state of a Gated Recurrent Unit (GRU, Cho et al., 2014) with state transition $\boldsymbol{d}_t = f([\boldsymbol{x}_{t-1}, \boldsymbol{z}_{t-1}], \boldsymbol{d}_{t-1}) \in \mathbb{R}^{D_d}$ where $[\cdot, \cdot]$ denotes concatenation. The joint and variational posterior distributions are given by

$$p(\boldsymbol{x}, \boldsymbol{z}) = \prod_{t=1}^{T} p(\boldsymbol{x}_t|\boldsymbol{x}_{<t}, \boldsymbol{z}_{\leq t})p(\boldsymbol{z}_t|\boldsymbol{x}_{<t}, \boldsymbol{z}_{<t}), \quad q(\boldsymbol{z}|\boldsymbol{x}) = \prod_{t=1}^{T} q(\boldsymbol{z}_t|\boldsymbol{x}_{\leq t}, \boldsymbol{z}_{<t}) \ . \quad (2)$$

**Stochastic recurrent neural network (SRNN).** The SRNN (Fraccaro et al., 2016) is similar to the VRNN and its joint can be written as in equation 2. Contrary to the VRNN, the SRNN has GRU state transitions that are independent of $\boldsymbol{z}_{1:T}$ such that $\boldsymbol{d}_t = f(\boldsymbol{x}_{t-1}, \boldsymbol{d}_{t-1})$. Furthermore, the SRNN conditions on the full observed sequence for inference, $q(\boldsymbol{z}_t|\boldsymbol{x}_{1:T}, \boldsymbol{z}_{t-1})$ via a second GRU with transition $\boldsymbol{a}_t = g([\boldsymbol{x}_t, \boldsymbol{d}_t], \boldsymbol{a}_{t+1})$. Then $\boldsymbol{z}_t$ is inferred from $\boldsymbol{a}_t$ and $\boldsymbol{z}_{t-1}$. This better approximates the true posterior which can be shown to depend on the full observed sequence (Bayer et al., 2021).

**Stochastic temporal convolutional network (STCN).** Contrary to VRNN and SRNN, the latent variables of the STCN are conditionally independent given $\boldsymbol{x}_{1:T}$ since there are no transition functions connecting them over time. Instead, a latent $\boldsymbol{z}_t^{(l)}$ at layer $l$ is conditioned on the latent variable above it $\boldsymbol{z}_t^{(l+1)}$ and a window of $\boldsymbol{x}_{1:T}$ defined by $\mathcal{R}_t^{(l)} = \{t - r_l + 1, \ldots, t\}$ via a WaveNet encoder with receptive field $r_l$ at layer $l$ (van den Oord et al., 2016). The joint and variational posterior are,

$$p(\boldsymbol{x}, \boldsymbol{z}) = \prod_{t=1}^{T} p(\boldsymbol{x}_t | \boldsymbol{z}_{\mathcal{R}_t}) \prod_{l=1}^{L} p(\boldsymbol{z}_t^{(l)} | \boldsymbol{x}_{\mathcal{R}_{t-1}^{(l)}}, \boldsymbol{z}_t^{(l+1)}) , \ \ q(\boldsymbol{z} | \boldsymbol{x}) = \prod_{t=1}^{T} \prod_{l=1}^{L} q(\boldsymbol{z}_t^{(l)} | \boldsymbol{x}_{\mathcal{R}_t^{(l)}}, \boldsymbol{z}_t^{(l+1)}) ,$$

where $\boldsymbol{z}_t^{(L+1)} := \emptyset$ and $\boldsymbol{z} = \boldsymbol{z}_{1:T}^{(1:L)}$ for notational convenience and inference is done top-down (Sønderby et al., 2016). The observation model $p(\boldsymbol{x}_t | \boldsymbol{z}_{\mathcal{R}_t^{(1)}}^{(1:L)})$ is also parameterized by a WaveNet.

**Clockwork variational autoencoder (CW-VAE).** The CW-VAE (Saxena et al., 2021) is a hierarchical LVM originally introduced for video generation. As seen in figure 1, it is autoregressive in the latent space but not in the observed space, contrary to the VRNN, SRNN and STCN. Additionally, each latent variable is updated only every $s_l$ timesteps, where $s_l$ is a layer-dependent stride and $s_1 < s_2 < \cdots < s_L$. We define $\mathcal{J}_t := \{l \,|\, t \in \mathcal{T}_l\}$ and $\mathcal{T}_l := \{t \in [1, T] \,|\, (t-1) \bmod s_l = 0\}$. Then,

$$p(\boldsymbol{x}, \boldsymbol{z}) = \prod_{t=1}^{T} p(\boldsymbol{x}_t | \boldsymbol{z}_t^{(1)}) \prod_{l \in \mathcal{J}_t} p(\boldsymbol{z}_t^{(l)} | \boldsymbol{z}_{t-s_l}^{(l)}, \boldsymbol{z}_t^{(l+1)}) , \ \ q(\boldsymbol{z} | \boldsymbol{x}) = \prod_{t=1}^{T} \prod_{l \in \mathcal{J}_t} q(\boldsymbol{z}_t^{(l)} | \boldsymbol{z}_{t-s_l}^{(l)}, \boldsymbol{z}_t^{(l+1)}, \boldsymbol{x}_{t:t+s_l}) .$$

The original encoder and decoder are not directly applicable to speech, since the sampling rates of speech are much higher than those of video (e.g. $16\,000\,\mathrm{Hz}$ compared to $30\,\mathrm{Hz}$). Hence, we use a convolutional ladder network similar to the STCN for inference and downsample the waveform.

**Output distribution.** Recent work on LVMs for speech modeling, including those considered here, often uses a Gaussian output distribution (Chung et al., 2015; Fraccaro et al., 2016; Hsu et al., 2017; Lai et al., 2018; Aksan & Hilliges, 2019; Zhu et al., 2020). Since audio is a naturally continuous signal, this may seem like an appropriate modeling choice. However, common audio datasets are sampled at bit-depths of $16\,\mathrm{bit}$ (TIMIT, Garofolo 1993; LibriSpeech, Panayotov et al. 2015). This results in a quantization gap between unique values of $1 \times 10^{-5}$ which increases the risk of an ill-posed problem by a likelihood that is unbounded from above unless the variance is lower bounded (Mattei & Frellsen, 2018). As a result, *reported likelihoods can be sensitive to hyperparameter settings and be hard to compare*. We discuss this phenomenon further in appendix H.

In this work, we therefore benchmark models using a discretized mixture of logistics (DMoL) as output distribution. The DMoL was introduced for image modeling with autoregressive models (Salimans et al., 2017) but has become standard in other generative models (Maaløe et al., 2019; Vahdat & Kautz, 2020; Child, 2021). Although continuous distributions can be used if the data is dequantized (Dinh et al., 2015; Theis et al., 2016; Ho et al., 2019), we do not consider this option here.

## 3 SPEECH MODELING BENCHMARK

**Data.** We train models on TIMIT (Garofolo, 1993) and randomly sample 5% of the training split for validation. Both input and target are $\mu$-law PCM standardized to $[-1, 1]$. We report on LibriSpeech (Panayotov et al., 2015) and linear PCM in appendix G. We further describe datasets in appendix C.

**Likelihood.** We report likelihoods in units of bits per frame (bpf; lower is better) as this yields a more interpretable likelihood that has connections to information theory and compression (Shannon, 1948; Townsend et al., 2019) compared to total likelihood in nats. For LVMs, we report the one-sample ELBO. The likelihoods can be seen in table 1. We describe how to convert to bpf in appendix F.

**Models.** We configure WaveNet, VRNN, SRNN, STCN and CW-VAE as in the original papers with the choices described here. Specifically, WaveNet uses ten layers, five blocks and $D_c = 96$ channels. VRNN and SRNN use latent dimension $D_z = 256$ and an equal number of hidden units. The STCN is in the dense configuration and uses 256 convolution channels, $L = 5$ layers and latent variables of dimensions $16, 32, 64, 128, 256$ from the top down. We also run a one-layered ablation with the same architecture but only one latent variable of dimension 256 at the top. The CW-VAE has $L = 1$ or $2$ latent variables of dimension 96 equal to the number of convolution channels and we let $s$ refer to $s_1$

| $s$ | Model | Configuration | $\mathcal{L}$ [bpf] |
|---|---|---|---|
| 1 | Uniform | Uninformed | 16.00 |
| 1 | DMoL | Optimal | 15.60 |
| 1 | WaveNet | $D_c = 96$ | **10.88** |
| 1 | LSTM | $D_d = 256$ | 10.97 |
| 1 | VRNN | $D_z = 256$ | $\leq$11.09 |
| 1 | SRNN | $D_z = 256$ | $\leq$11.19 |
| 1 | STCN | $D_z = 256, L = 5$ | $\leq$11.77 |
| 64 | WaveNet | $D_c = 96$ | 13.30 |
| 64 | LSTM | $D_d = 256$ | 13.34 |
| 64 | VRNN | $D_z = 256$ | $\leq$12.54 |
| 64 | SRNN | $D_z = 256$ | $\leq$12.42 |
| 64 | CW-VAE | $D_z = 96, L = 1$ | $\leq$12.44 |
| 64 | CW-VAE | $D_z = 96, L = 2$ | $\leq$12.17 |
| 64 | STCN | $D_z = 256, L = 1$ | $\leq$12.32 |
| 64 | STCN | $D_z = 256, L = 5$ | $\leq$**11.78** |

Table 1

| ASR configuration | | | Result |
|---|---|---|---|
| Data | Model | Input | PER [%] |
| 3.7h | Spectrogram | Mel | 24.1 |
| 3.7h | WaveNet | $h^{(15)}$ | 27.7 |
| 3.7h | LSTM | $h$ | 23.0 |
| 3.7h | VRNN | $z$ | 23.2 |
| 3.7h | SRNN | $z$ | 26.0 |
| 3.7h | CW-VAE | $z^{(1)}$ | 36.4 |
| 3.7h | STCN | $z^{(2)}$ | **21.9** |
| 1.0h | Spectrogram | Mel | 30.8 |
| 1.0h | WaveNet | $h^{(15)}$ | 34.7 |
| 1.0h | LSTM | $h$ | 30.1 |
| 1.0h | VRNN | $z$ | 30.4 |
| 1.0h | SRNN | $z$ | 31.7 |
| 1.0h | CW-VAE | $z^{(1)}$ | 40.0 |
| 1.0h | STCN | $z^{(2)}$ | **26.7** |

Table 2

In table 1 we report model likelihoods $\mathcal{L}$ for TIMIT represented as a 16bit $\mu$-law encoded PCM for different stochastic latent variable models and deterministic autoregressive baselines. In table 2 we report phoneme error rate (PER) of different representations for phoneme recognition on TIMIT.

and set $s_2 = 8s_1$. We also train an LSTM baseline (Hochreiter & Schmidhuber, 1997) configured similar to the VRNN. All models use a 10 component DMoL output distribution. All LVMs use diagonal covariance Gaussian priors and posteriors. We train and evaluate models with waveforms stacked similar to previous work (Chung et al., 2015; Fraccaro et al., 2016; Aksan & Hilliges, 2019) with stack sizes of $s = 1$, $s = 64$ and $s = 256$. Finally, we evaluate a per-frame discrete uniform distribution and a two-component DMoL fitted to the training set to estimate worst case performance. We further describe models and training in appendix D and E and provide results with a Gaussian output distribution in appendix G.

**Results.** We present the model likelihoods for $s = 1$ and $64$ in table 1 and for $s = 256$ in appendix G. It is clear that deterministic autoregressive models are superior to LVMs at $s = 1$, but inferior to them at $s = 64$ and $256$. For strides $s > 1$, previous work has attributed the inferior performance of autoregressive models without latent variables to the ability of LVMs to model intra-step correlations (Lai et al., 2019). It should be noted that at $s = 1$, the STCN was numerically unstable and that the CW-VAE was computationally infeasible to train. Surprisingly, the simple LSTM performs almost on par with WaveNet. The autoregressive STCN is the best-performing LVM. For both CW-VAE and STCN, increasing the number of latent variables in the hierarchy improves performance, similar to results in the image domain. The best performing LVMs, STCN and CW-VAE, are not yet scalable to $s = 1$ resolution where the highest likelihoods are achieved. Hence, LVMs may be able to outperform autoregressive models at $s = 1$ in the future.

**Phoneme recognition.** Although the likelihood is a practical metric for model comparison, a high likelihood does not guarantee that a model has learned useful representations (Huszár, 2017). To assess the usefulness of the representations, we train a model to recognize phonemes on the TIMIT data. We compare the phoneme error rate (PER) of the model when using input representations obtained from different unsupervised models, trained on the full TIMIT dataset (3.7h) at $s = 64$, as in table 1, and when using a Mel spectrogram (80 filterbanks, hop length 64, window size 128). The recognition model is a three-layered bidirectional LSTM with 256 hidden units. It is trained with the connectionist temporal classification (CTC) loss (Graves et al., 2006). We report the PER in table 2.

As expected, Mel spectrograms perform well achieving 24.1% PER using 3.7 hours of labeled data. However, the ASR trained on STCN representations outperforms the Mel spectrogram with a PER of 21.9%. This indicates that unsupervised STCN representations are phonetically rich and potentially better suited for speech modeling than the engineered Mel spectrogram. When the amount of labeled data is reduced, LVM representations suffer slightly less than deterministic ones. Interestingly, representations from WaveNet are outperformed by all LVMs and the LSTM.

ACKNOWLEDGMENTS

This research was partially funded by the Innovation Fund Denmark via the Industrial PhD Programme (grant no. 0153-00167B). JF and SH were funded in part by the Novo Nordisk Foundation (grant no. NNF20OC0062606) via the Center for Basic Machine Learning Research in Life Science (MLLS, https://www.mlls.dk). JF was further funded by the Novo Nordisk Foundation (grant no. NNF20OC0065611) and the Independent Research Fund Denmark (grant no. 9131-00082B). SH was further funded by VILLUM FONDEN (15334) and the European Research Council (ERC) under the European Union's Horizon 2020 research and innovation programme (grant agreement no. 757360).

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

## A    REPRODUCIBILITY STATEMENT

The source code used for the work presented in this paper will be made available before the conference. This code provides all details, practical and otherwise, needed to reproduce the results in this paper including data preprocessing, model training, model likelihood and latent space evaluation. The source code also includes scripts for downloading and preparing the LibriSpeech, LibriLight and TIMIT datasets. The LibriSpeech and LibriLight datasets are open source and can be downloaded with the preparation scripts. They are also available at `https://www.openslr.org/12` and `https://github.com/facebookresearch/libri-light`, respectively. The TIMIT dataset is commercial and must be purchased and downloaded from `https://catalog.ldc.upenn.edu/LDC93S1` before running the preparation script.

The stochastic latent variable models considered in this work do not provide an exact likelihood estimate nor an exact latent space representation. For the likelihood, they provide a stochastic lower bound and some variation in the reproduced likelihoods as well as latent representations must be expected between otherwise completely identical forward passes. This variance is fairly small in practice when averaging over large datasets such as those considered in this work. We seed our experiments to reduce the randomness to a minimum, but parts of the algorithms underlying the CUDA framework are stochastic for efficiency. To retain computational feasibility, we do not run experiments with a deterministic CUDA backend.

## B    ETHICS STATEMENT

The work presented here fundamentally deals with automated perception of speech and generation of speech. These applications of machine learning potentially raise a number of ethical concerns. For instance, the these models might see possibly adverse use in automated surveillance and generation of deep fakes. To counter some of these effects, this work has focused on openness by using publicly available datasets for model development and benchmarking. Additionally, the work will open source the source code used to create these results. Ensuring the net positive effect of the development of these technologies is and must continue to be an ongoing effort.

We do not associate any significant ethical concerns with the datasets used in this work. However, one might note that the TIMIT dataset has somewhat skewed distributions in terms of gender and race diversity. Specifically, the male to female ratio is about two to one while the vast majority of speakers are Caucasian. Such statistics might have an effect of some ethical concern on downstream applications derived from such a dataset as also highlighted in recent research (Koenecke et al., 2020). In LibriSpeech, there is an approximately equal number of female and male speakers while the diversity in race is unknown to the authors.

## C    DATASETS

**TIMIT.** TIMIT (Garofolo, 1993) is a speech dataset which contains $16\,\text{kHz}$ recordings of 630 speakers of eight major dialects of American English, each reading ten phonetically rich sentences. It amounts to 6300 total recordings splits approximately in 3.94 hours of audio for training and 1.43 hours of audio for testing. No speakers or sentences in the test set are in the training set. The full train and test subsets of TIMIT are as in previous work (Chung et al., 2015; Fraccaro et al., 2016; Aksan & Hilliges, 2019). We randomly sample 5% of the training set to use as a validation set. TIMIT includes temporally aligned annotations of phonemes and words as well as speaker metadata such as gender, height, age, race, education level and dialect region (Garofolo, 1993).

**LibriSpeech and LibriLight.** The LibriSpeech dataset (Panayotov et al., 2015) consists of readings of public domain audio books amounting to approximately $1000\,\text{h}$ of audio. The data is derived from the LibriVox project. LibriLight (Kahn et al., 2020) is a subset of LibriSpeech created as an automatic speech transcription (ASR) benchmark with limited or no supervision. We specifically train on the $100\,\text{h}$ train-clean-100 subset of LibriSpeech and the $10\,\text{h}$ subset of LibriLight. In all cases we evaluate on all the test splits dev-clean, dev-other, test-clean, test-other.

Both datasets represent the audio as $16\,\text{bit}$ pulse code modulation (PCM) sampled at $16\,000\,\text{Hz}$.

## D    MODEL ARCHITECTURES

This section details model architectures. See appendix J for graphical models and appendix E for training details.

**WaveNet.**   We implement WaveNet as described in the original work (van den Oord et al., 2016) but use a discretized mixture of logistics as the output distribution as also done in other work (van den Oord et al., 2018a). Our WaveNet is not conditioned on any signal other than the raw waveform. The model applies the causal convolution directly to the raw waveform frames (i.e. one input channel). An alternative option that we did not examine is to replace the initial convolution with an embedding lookup with a learnable vector for each waveform frame value.

**LSTM.**   The LSTM baseline uses an MLP encoder to embed the waveform subsegment $x_{t:t+s-1}$ to a feature vector before feeding it to the LSTM cell. The encoder is similar to the parameterization of $\phi_{\text{vrnn}}^{\text{enc}}$ for the VRNN described above. The LSTM cell produces the hidden state $d_t$ from $x_{t:t+s-1}$ and passes it to a decoder. Like the encoder, the decoder is parameterized like $\phi_{\text{vrnn}}^{\text{dec}}$ of the VRNN. It outputs the waveform predictions $x_{t+s:t+2s-1}$ from the hidden state $d_t$. The LSTM model uses a single vanilla unidirectional LSTM cell.

**VRNN.**   We implement the VRNN as described in the original work (Chung et al., 2015) and verify that we can reproduce the original Gaussian likelihood TIMIT results. We replace the Gaussian output distribution with the DMoL.

**SRNN.**   We implement the VRNN as described in the original work (Fraccaro et al., 2016) and verify that we can reproduce the original Gaussian likelihood TIMIT results. We replace the Gaussian output distribution with the DMoL.

**CW-VAE.**   We implement the CW-VAE based on the original work (Saxena et al., 2021) but with some modifications also briefly described in the paper. We replace the encoder/decoder model architectures of the original work with architectures designed for waveform modeling. Specifically, the encoder and decoder are based on the Conv-TasNet (Luo & Mesgarani, 2019) and uses similar residual block structure. However, contrary to the Conv-TasNet, we require downsampling factors larger than two. In order to achieve this we use strides of two in the separable convolution of each block. With e.g. six blocks we hence get an overall stride of $2^6 = 64$. We can then add additional blocks with unit stride. We also need to modify the residual connections that skip strided convolutions. Specifically, we replace the residual with a single convolution with stride equal to the stride used in the separable convolution. This convolution uses no nonlinearity and hence simply learns a local linear downsampling.

**STCN.**   We implement the STCN as described in the original work (Aksan & Hilliges, 2019) and verify that we can reproduce the original Gaussian likelihood TIMIT results. We replace the Gaussian output distribution with the DMoL. We use the best-performing version of the STCN reported in the original paper, namely the "STCN-dense" variant which conditions the observed variable on all five latent variables in the hierarchy. For the ablation experiment, we remove the bottom four latent variables. That is, we completely remove the corresponding four small densely connected networks that parameterize the prior and posterior distributions based on deterministic representations of the WaveNet encoder. We keep the top most prior and posterior networks and use them to parameterize a latent variable of 256. This maintains the widest bottleneck of the model as well as almost all of the model's capacity.

**ASR model.**   The ASR model used for the phoneme recognition experiments is a three-layered bidirectional LSTM. We apply temporal dropout between the LSTM layers and also after the final layer. Temporal dropout works similar to regular dropout but samples the entries of the hidden state to mask only once and apply it to all timesteps, i.e. masking $h_t$ at vector index $i$ for all $t$ (and $i$). We mask by zeroing vector elements. We never mask the first timestep. We apply temporal dropout with masking probability of 0.3 for the 3.7h subset, 0.35 for the 1h subset and 0.4 for the 10m subset. The only difference in model architecture between the evaluation of different representations is the first affine transformation; from the dimensionality of the representation to the hidden state size of the LSTM. This gives rise to a very small difference in model capacity and parameter count which we find is negligible. We set the hidden unit size to 256.

# E   TRAINING DETAILS

**Likelihood benchmark.**   We implement all models and training scripts in PyTorch 1.9 (Paszke et al., 2017). For both datasets we use the Adam optimizer (Kingma & Ba, 2015) with default parameters as given in PyTorch. We use learning rate $3e - 4$ and no learning rate schedule. We use PyTorch automatic mixed precision (AMP) to significantly reduce memory consumption. We did not observe any significant difference in final model performance compared to full ($32 \, \text{bit}$) precision.

We train stateful models (LSTM, VRNN, SRNN and CW-VAE) on the full sequence lengths padding batches with zeros when examples are not of equal length. We sample batches such that they consist of examples that are approximately the same length to minimize the amount of computation wasted on padding.

For $s = 1$, we train stateless models (WaveNet, STCN) on random subsegments of the training examples and resample every epoch. This reduces memory requirements but does not bias the gradient. The subsequences are chosen to be of length 16000 which is larger than the receptive fields of the models and corresponds to one second of audio in TIMIT and LibriSpeech. For $s = 64$ and $s = 256$ we train the stateless models on the full example lengths similar to the stateful models since the receptive field is effectively $s$ times larger and the shorter sequence length reduces memory requirements.

In testing, we evaluate on the full sequences. Due to memory constrains, for LibriSpeech, we need to split the test examples into subsegments since the average sequence length in Librispeech is about 4 times longer than that of TIMIT. Hence, we do multiple forward passes per test example, one for each of several subsegments. We carry along the internal state for models that are autoregressive in training (LSTM, VRNN, SRNN, CW-VAE) and define segments to overlap according to model architecture.

**Phoneme recognition.**   The ASR experiment consists of two stages: 1) pre-training of the unsupervised model and 2) training of the ASR model. The pre-training is done as for the likelihood benchmark above. The ASR model is trained using the Adam optimizer (Kingma & Ba, 2015) with default parameters as given in PyTorch. We use learning rate $3e - 4$ and no learning rate schedule.

For the spectrogram, WaveNet and the LSTM, we extract the representation only once and train the ASR model on these. Since the models are deterministic and do not parameterize distributions, this is the only option. For the LVMs, we resample the latent representation of a training example at every epoch. This is the most principled approach as these models parameterize probability distributions. Furthermore, using a single sample would be subject to artificially high variance in the representations while it is not straightforward to establish a sound mean representation for sequential models.

# F   CONVERTING THE LIKELIHOOD TO UNITS OF BITS PER FRAME

Here we briefly describe how to compute a likelihood in units of bits per frame (bpf). In the main text, we use $\log$ to mean $\log_e$, but here we will be explicit. In general, conversion from nats to bits (i.e., from $\log_e$ to $\log_2$) is achieved by $\log_2(x) = \log_e(x) / \log_2(e)$. Remember that $\log_2 p(\boldsymbol{x}_{1:T})$ generally factorizes as $\sum_t \log_2 p(x_t | \cdot)$. In sequence modeling, it is important to remember that each example $\boldsymbol{x}^i$ must be weighted differently according the sequence length of that specific example. This is in contrast to computing bits per dimension in the image domain where images in a dataset are usually of the same dimensions. Thus, we compute the log-likelihood in bits per frame over the entire dataset as

$$\mathcal{L}(\boldsymbol{x}^i) = \frac{1}{\sum_i T_i} \sum_i \sum_t \log_2 p(\boldsymbol{x}_t^i) \;, \tag{3}$$

where $i$ denotes the example index, $T_i$ is the length of example $\boldsymbol{x}^i$ in waveform frames and $t$ is the time index. If a single timestep $\boldsymbol{x}_t^i$ represents multiple waveform frames stacked with some stack size $s$, it is important to note that the sum over $t$ only has $T_i / s$ elements. For the LVMs, the term $\log_2 p(\boldsymbol{x}_t^i)$ is lower bounded by the ELBO in equation 1.

| $s$ | Model | Configuration | Likelihood $\mathcal{L}$ [bpf] | | | |
|---|---|---|---|---|---|---|
| | | | dev-clean 10h/100h | dev-other 10h/100h | test-clean 10h/100h | test-other 10h/100h |
| 1 | Uniform | Uninformed | 16.00 | 16.00 | 16.00 | 16.00 |
| 1 | DMoL | Optimal | 15.66 | 15.70 | 15.62 | 15.71 |
| - | FLAC | Linear PCM | **9.390** | **9.292** | **9.700** | **9.272** |
| 1 | Wavenet | $D_c = 96$ | **10.96/10.89** | **10.85/10.76** | **11.12/11.01** | **11.05/10.85** |
| 1 | LSTM | $D_d = 256$ | 11.21/11.17 | 11.10/11.06 | 11.35/11.29 | 11.28/11.23 |
| 64 | Wavenet | $D_c = 96$ | 13.61/13.24 | 13.58/13.21 | 13.61/13.22 | 13.60/13.21 |
| 64 | LSTM | $D_d = 256$ | 13.56/13.25 | 13.52/13.24 | 13.55/13.23 | 13.56/13.25 |
| 64 | CW-VAE | $D_z = 96, L = 1$ | $\leq$12.32/12.24 | 12.32/12.23 | 12.43/12.33 | 12.43/12.33 |
| 64 | CW-VAE | $D_z = 96, L = 2$ | $\leq$12.30/12.22 | 12.30/12.21 | 12.40/12.31 | 12.39/12.32 |
| 64 | STCN | $D_z = 256, L = 5$ | $\leq$**11.83/11.47** | **11.82/11.46** | **11.94/11.58** | **11.94/11.60** |

Table 3: Model likelihoods $\mathcal{L}$ on **LibriSpeech** test sets represented as **16 bit $\mu$-law encoded PCM**. For the CW-VAE, $s$ refers to $s_1$ and the two-layered models have $s_2 = 8s_1$. The models are trained on either the $10\,\mathrm{h}$ LibriLight subset or the $100\,\mathrm{h}$ LibriSpeech train-clean-100 subset as indicated. Likelihoods are given in units of bits per frame (bpf).

## G  ADDITIONAL LIKELIHOOD RESULTS

**LibriSpeech, $\mu$-law, DMoL.**  We provide additional results on LibriSpeech with audio represented as $\mu$-law encoded PCM in table 3. See appendix C, D and E for additional details.

**TIMIT, $\mu$-law, DMoL.**  We provide additional results on TIMIT with audio represented as $\mu$-law encoded PCM in table 5. Details are as presented in the main paper.

**TIMIT, linear, DMoL.**  : We provide results on TIMIT with audio represented as linear PCM (raw PCM) in table 4. Except for the encoding, details are as for $\mu$-law encoded TIMIT

**TIMIT, linear, Gaussian.**  We also provide some results on TIMIT with the audio instead represented as linear PCM (linearly encoded) and using Gaussian output distributions as has been done previously in the literature (Chung et al., 2015; Fraccaro et al., 2016; Lai et al., 2018; Aksan & Hilliges, 2019). We use $s = 200$ for comparability to the previous work. We provide the results in table 6 and include likelihoods reported in the literature for reference. For our models, we use the same architectures as before but replace the discretized mixture of logistics with either a Gaussian distribution or a mixture of Gaussian distributions.

We constrain the standard deviation of the Gaussians used with our models to be at least $\sigma_{\min} = 0.01$ in order to avoid it going to zero, the likelihood going to infinity and optimization becoming unstable. The minimum Gaussian standard deviation of Aksan & Hilliges (2019) is $\sigma_{\min} = 0.001$.

From table 6 we note that the performance of the CW-VAE with Gaussian output distribution when modeling linear PCM (i.e. not $\mu$-law encoded) does not compare as favorably to the other baselines as it did with the discretized mixture of logistics distribution. We hypothesize that this has to do with using a Gaussian output distribution in latent variable models which, as has been reported elsewhere (Mattei & Frellsen, 2018), leads to a likelihood function that is unbounded above and can grow arbitrarily high. We discuss this phenomenon in further detail in section H.

We specifically hypothesize that models that are autoregressive in the observed variable (VRNN, SRNN, Stochastic WaveNet, STCN) are well-equipped to utilize local smoothness to put very high density on the correct next value and that this in turn leads to a high degree of exploitation of the unboundedness of the likelihood. Not being autoregressive in the observed variable, the CW-VAE cannot exploit this local smoothness in the same way. Instead, the reconstruction is conditioned on a stochastic latent variable, $p(\boldsymbol{x}_t|\boldsymbol{z}_t^1)$, which introduces uncertainty and likely larger reconstruction variances.

| $s$ | Model | Configuration | $\mathcal{L}$ [bpf] |
|---|---|---|---|
| 1 | Uniform | Uninformed | 16.00 |
| 1 | DMoL | Optimal | 10.70 |
| 1 | Wavenet | $D_C = 96$ | **7.246** |
| 1 | LSTM | $D_d = 256, L = 1$ | 7.295 |
| 1 | VRNN | $D_z = 256$ | $\leq 7.316$ |
| 1 | SRNN | $D_z = 256$ | $\leq 7.501$ |
| 1 | STCN | $D_z = 256, L = 5$ | $\leq 9.970$ |
| 64 | WaveNet | $D_c = 96$ | 8.402 |
| 64 | LSTM | $D_d = 256, L = 1$ | 8.357 |
| 64 | VRNN | $D_z = 256$ | $\leq 8.103$ |
| 64 | SRNN | $D_z = 256$ | $\leq 8.036$ |
| 64 | CW-VAE | $D_z = 96, L = 1$ | $\leq 7.989$ |
| 64 | STCN | $D_z = 256, L = 5$ | $\leq$**7.768** |
| 256 | WaveNet | $D_c = 96$ | 9.018 |
| 256 | LSTM | $D_d = 256, L = 1$ | 8.959 |
| 256 | VRNN | $D_z = 256$ | $\leq 8.739$ |
| 256 | SRNN | $D_z = 256$ | $\leq 8.674$ |
| 256 | CW-VAE | $D_z = 96, L = 1$ | $\leq 8.406$ |
| 256 | STCN | $D_z = 256, L = 5$ | $\leq$**8.196** |

Table 4: Model likelihoods on **TIMIT** represented as a **16 bit linear PCM**, obtained by different latent variable models and compared to autoregressive baselines all using a discretized mixture of logistics with 10 components as output distribution. Likelihoods are given in units of bits per frame (bpf) and obtained by normalizing the total likelihood of each sequence with the individual sequence length and then averaging over the dataset. The STCN converges to a poor local minimum and sometimes diverges when modeling linear PCM with $s = 1$.

## H    ADDITIONAL DISCUSSION ON GAUSSIAN LIKELIHOODS IN LVMS

As noted in section G, we constrain the variance of the output distribution of our models to be $\sigma^2_{\min} = 0.01^2$ for the additional results on TIMIT with Gaussian outputs. This limits the maximum value attainable by the prediction/reconstruction density of a single waveform frame $x_t$.

Specifically, we can see that since

$$\log p(x_t | \cdot) = \log \mathcal{N}\big(x_t; \mu_t, \max\big\{\sigma^2_{\min}, \sigma^2_t\big\}\big) \ , \tag{4}$$

the best prediction/reconstruction density is achieved when $\sigma^2 \leq \sigma^2_{\min}$ and $\mu = x_t$. Here $\cdot$ indicates any variables we might condition on such as the previous input frame $x_{t-1}$ or some latent variables. We can evaluate this best case scenario for $\sigma^2_{\min} = 0.01^2$,

$$\begin{aligned}
\log \mathcal{N}\big(x_t; x_t, \sigma^2_{\min}\big) &= -\frac{1}{2}\log 2\pi - \frac{1}{2}\log \sigma^2_{\min} - \frac{1}{2\sigma^2_{\min}}(x_t - x_t) \\
&= -\frac{1}{2}\log 2\pi - \frac{1}{2}\log 0.01^2 \\
&= 3.686 \ . 
\end{aligned} \tag{5}$$

Hence, with perfect prediction/reconstruction and the minimal variance $(0.01^2)$, a waveform frame contributes to the likelihood with $3.686 \, \text{nats}$. With an average test set example length of $49\,367.3 \, \text{frames}$ frames this leads to a best-case likelihood of $181967$. We provide a list of maximally attainable Gaussian likelihoods on TIMIT for different minimal variances in table 7. One can note that the maximal likelihood at $\sigma^2_{\min} = 0.1^2$ is lower than the likelihoods achieved by some models in table 6. This indicates that the models learn to use very small variances in order to increase the likelihood. Empirically, standard deviations smaller than approximately 0.001 can result in numerical instability.

| $s$ | Model | Configuration | $\mathcal{L}$ [bpf] |
|---|---|---|---|
| 1 | Wavenet | $D_C = 16$ | 11.27 |
| 1 | Wavenet | $D_C = 24$ | 11.14 |
| 1 | Wavenet | $D_C = 32$ | 11.03 |
| 1 | Wavenet | $D_C = 96$ | 10.88 |
| 1 | Wavenet | $D_C = 128$ | 10.98 |
| 1 | Wavenet | $D_C = 160$ | 10.91 |
| 1 | LSTM | $D_d = 128, L = 1$ | 11.40 |
| 1 | LSTM | $D_d = 256, L = 1$ | 11.11 |
| 1 | VRNN | $D_z = 256$ | $\leq 11.09$ |
| 1 | SRNN | $D_z = 256$ | $\leq 11.19$ |
| 1 | STCN | $D_z = 256, L = 5$ | $\leq 11.77$ |
| 4 | LSTM | $D_d = 256, L = 1$ | 11.65 |
| 16 | LSTM | $D_d = 256, L = 1$ | 12.54 |
| 16 | LSTM | $D_d = 256, L = 2$ | 12.54 |
| 16 | LSTM | $D_d = 256, L = 3$ | 12.44 |
| 64 | WaveNet | $D_c = 96$ | 13.30 |
| 64 | LSTM | $D_d = 96, L = 1$ | 13.49 |
| 64 | LSTM | $D_d = 96, L = 2$ | 13.46 |
| 64 | LSTM | $D_d = 96, L = 3$ | 13.40 |
| 64 | LSTM | $D_d = 256, L = 1$ | 13.27 |
| 64 | LSTM | $D_d = 256, L = 2$ | 13.29 |
| 64 | LSTM | $D_d = 256, L = 3$ | 13.31 |
| 64 | LSTM | $D_d = 512, L = 1$ | 13.37 |
| 64 | LSTM | $D_d = 512, L = 2$ | 13.37 |
| 64 | LSTM | $D_d = 512, L = 3$ | 13.41 |
| 64 | VRNN | $D_z = 96$ | $\leq 12.93$ |
| 64 | VRNN | $D_z = 256$ | $\leq 12.54$ |
| 64 | SRNN | $D_z = 96$ | $\leq 12.87$ |
| 64 | SRNN | $D_z = 256$ | $\leq 12.42$ |
| 64 | CW-VAE | $D_z = 96, L = 1$ | $\leq 12.44$ |
| 64 | CW-VAE | $D_z = 96, L = 2$ | $\leq 12.17$ |
| 64 | CW-VAE | $D_z = 96, L = 3$ | $\leq 12.15$ |
| 64 | CW-VAE | $D_z = 256, L = 2$ | $\leq 12.10$ |
| 64 | STCN | $D_z = 256, L = 1$ | $\leq 12.32$ |
| 64 | STCN | $D_z = 256, L = 5$ | $\leq 11.78$ |
| 256 | WaveNet | $D_c = 96$ | 14.11 |
| 256 | LSTM | $D_d = 256, L = 1$ | 14.20 |
| 256 | LSTM | $D_d = 256, L = 2$ | 14.17 |
| 256 | LSTM | $D_d = 256, L = 3$ | 14.26 |
| 256 | VRNN | $D_z = 96$ | $\leq 13.51$ |
| 256 | VRNN | $D_z = 256$ | $\leq 13.27$ |
| 256 | SRNN | $D_z = 96$ | $\leq 13.28$ |
| 256 | SRNN | $D_z = 256$ | $\leq 13.14$ |
| 256 | CW-VAE | $D_z = 96, L = 1$ | $\leq 13.11$ |
| 256 | CW-VAE | $D_z = 96, L = 2$ | $\leq 12.97$ |
| 256 | CW-VAE | $D_z = 96, L = 3$ | $\leq 12.87$ |
| 256 | STCN | $D_z = 256, L = 1$ | $\leq 13.07$ |
| 256 | STCN | $D_z = 256, L = 5$ | $\leq 12.52$ |

Table 5: Model likelihoods on **TIMIT** represented as a **16 bit $\mu$-law encoded PCM**, obtained by different latent variable models and compared to autoregressive baselines all using a discretized mixture of logistics with 10 components as output distribution. Likelihoods are given in units of bits per frame (bpf) and obtained by normalizing the total likelihood of each sequence with the individual sequence length and then averaging over the dataset.

| $s$ | Model | Configuration | $\mathcal{L}$ [nats] |
|---|---|---|---|
| 1 | WaveNet | Normal | 119656 |
| 1 | WaveNet | GMM-2 | 120699 |
| 1 | WaveNet | GMM-20 | 121681 |
| 200 | WaveNet (Aksan & Hilliges, 2019) | GMM-20 | 30188 |
| 200 | WaveNet (Aksan & Hilliges, 2019) | Normal | -7443 |
| 200 | Stochastic WaveNet* (Lai et al., 2018) | Normal | $\geq$72463 |
| 200 | VRNN (Chung et al., 2015) | Normal | $\approx$28982 |
| 200 | SRNN (Fraccaro et al., 2016) | Normal | $\geq$60550 |
| 200 | STCN (Aksan & Hilliges, 2019) | GMM-20 | $\geq$69195 |
| 200 | STCN (Aksan & Hilliges, 2019) | Normal | $\geq$64913 |
| 200 | STCN-dense (Aksan & Hilliges, 2019) | GMM-20 | $\geq$71386 |
| 200 | STCN-dense (Aksan & Hilliges, 2019) | Normal | $\geq$70294 |
| 200 | STCN-dense-large (Aksan & Hilliges, 2019) | GMM-20 | $\geq$77438 |
| 200 | CW-VAE* | $L = 1, D_z = 96$, Normal | $\geq$41629 |

Table 6: Model likelihoods on **TIMIT** represented as **globally normalized 16 bit linear PCM**. Likelihoods are given in units of nats and obtained by summing the likelihood over time and over all examples in the dataset and dividing by the total number of examples. In the table, Normal refers to using a Gaussian likelihood and GMM refers to using a Gaussian Mixture Model likelihood with 20 components. Models with asterisks * are our implementations while remaining results are as reported in the referenced work.

| $\sigma_{\min}$ | $\sigma_{\min}^2$ | $\max \mathcal{L}$ |
|---|---|---|
| 1 | 1 | $-45367$ |
| 0.5 | 0.25 | $-11146$ |
| 0.1 | 0.01 | 68307 |
| 0.05 | 0.0025 | 102525 |
| 0.01 | 0.0001 | 181979 |
| 0.005 | 0.000025 | 216198 |
| 0.001 | 0.000001 | 295651 |

Table 7: The highest possible Gaussian log-likelihoods ($\max \mathcal{L}$) attainable on the TIMIT test set as computed by equation 4 with different values of the minimum variance $\sigma_{\min}^2$.

# I    ADDITIONAL DISCUSSION ON THE CHOICE OF OUTPUT DISTRIBUTION

The DMoL uses a discretization of the continuous logistic distribution to define a mixture model over a discrete random variable. This allows it to parameterize multimodal distributions which can express ambiguity about the value of $x_t$. The model can learn to maximize likelihood by assigning a bit of probability mass to multiple potential values of $x_t$.

While this is well-suited for autoregressive modeling, for which the distribution was developed, the potential multimodality poses a challenge for non-autoregressive latent variable models which independently sample multiple neighboring observations at the output. In fact, if multiple neighboring outputs defined by the subsequence $x_{t_1:t_2}$ have multimodal $p(x_t|\cdot)$, we risk sampling a subsequence where each neighboring value expresses different potential realities, independently.

Interestingly, most work on latent variable models with non-autoregressive output distributions seem to ignore this fact and simply employ the mixture distribution with 10 mixture components (Maaløe et al., 2019; Vahdat & Kautz, 2020; Child, 2021). However, given the empirically good results of latent variable models for image generation, this seems to have posed only a minor problem in practice. We speculate that this is due to the high degree of similarity between neighbouring pixels in images. I.e. if the neighboring pixels are nuances of red, then, in all likelihood, so is the central pixel.

In the audio domain, however, neighbouring waveform frames can take wildly different values, especially at low sample rates. Furthermore, waveforms exhibit a natural symmetry between positive

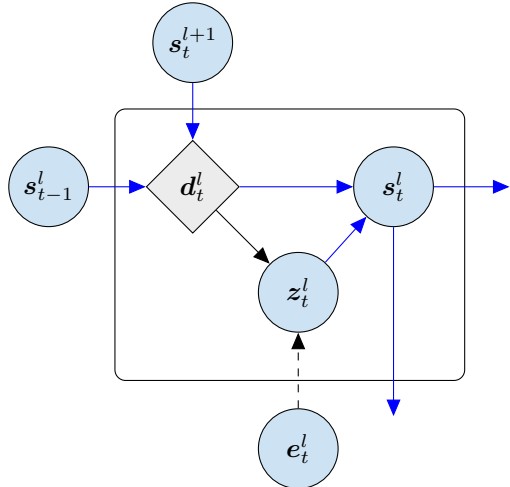

Figure 2: CW-VAE cell state $s_t^l$ update. The cell state is given as $s_t^l = (z_t^l, d_t^l)$ where $d_t^l$ is the deterministic hidden state of a Gated Recurrent Unit (Cho et al., 2014). The vector $e_t^l$ is computed from $x_t$ by the encoder network which outputs $L$ encodings, one for each latent variable, similar to that of a Ladder VAE Sønderby et al. (2016). All blue arrows are shared between generation and inference. The dashed arrow is used only during inference. The solid arrow has unique transformations during inference and generation.

and negative amplitudes. Hence, it seems plausible that multimodality may pose a larger problem in non-autoregressive speech generation by causing locally incoherent samples than it seems to do in image modelling.

## J    ADDITIONAL GRAPHICAL MODELS

In figure 2 we show the graphical model of the recurrent cell of the CW-VAE for a single time step. As noted in (Saxena et al., 2021), this cell is very similar to the one of the Recurrent State Space Model (RSSM) (Hafner et al., 2019).

In figure 3 we show the unrolled graphical models of a three-layered CW-VAE with $k_1 = 1$ and $c = 2$ yielding $k_2 = 2$ and $k_3 = 4$. We show both the generative and inference models and highlight in blue the parameter sharing between the two models due to top-down inference.

In figure 4 we show the graphical models of the STCN Aksan & Hilliges (2019) at a single timestep. The model has three layers and shares the parameters of the WaveNet encoder between the inference and generative models.

In figure 5 we illustrate the unrolled graphical models of the inference and generative models of the VRNN (Chung et al., 2015). We include the deterministic variable $d_t$ in order to illustrate the difference to other latent variable models.

Likewise, in figure 6 we illustrate the unrolled graphical models the SRNN (Fraccaro et al., 2016).

## K    ADDITIONAL LATENT EVALUATION

Here we present additional qualitative assesessment of the learned latent representations selectively for the CW-VAE.

In figure 7 we present evaluation in terms of phoneme clustering. Specifically, we infer the latent variables of all utterances by a single speaker from the TIMIT test set. We sample the latent sequence 100 times to estimate the mean representation per time step. We then compute the average latent representation over the duration of each phoneme using aligned phoneme labels. This approximately marginalizes out variation during the phoneme. We use linear discriminant analysis (LDA) Fisher

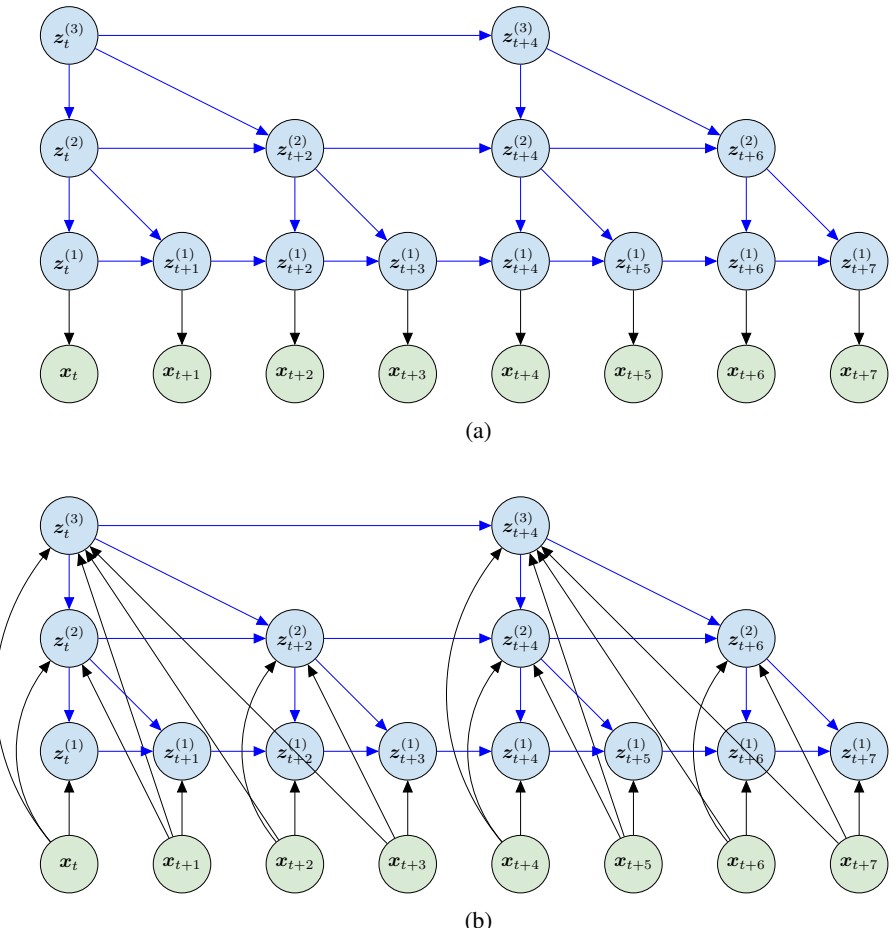

Figure 3: CW-VAE (Saxena et al., 2021) generative model $p(\boldsymbol{x}, \boldsymbol{z})$ in (a) and inference model $q(\boldsymbol{z}|\boldsymbol{x})$ in (b) for a three-layered model with $k_1 = 1$ and $c = 2$ giving $k_2 = 2$ and $k_3 = 4$ unrolled over eight steps in the observed variable. Blue arrows are (mostly) shared between the inference and generative models. See figure 2 for a detailed graphical model expanding on the latent nodes $\boldsymbol{z}_t^l$ and parameter sharing.

(1936) to obtain a low-dimensional linear subspace of the latent space. We visualize the resulting representations colored according to test set phoneme classes in figure 7. In the left plot, many phonemes are separable in the linear subspace and that related phonemes such as "s" and "sh" are close. In the right plot, we show the average accuracy of a leave-one-out $k$-nearest-neighbor (KNN) classifier on the single left-out latent representation reduced with a 5-dimensional LDA as a function of the number of neighbors. We compare accuracy to a Mel-spectrogram averaged over each phoneme duration and LDA reduced. The spectrogram is computed with hop length set to 64, equal to $s_1$ for the CW-VAE, window size 256 and 80 Mel bins. We see that both latent spaces yield significantly better KNN accuracies than the Mel features.

We visualize the performance of a $k$-nearest-neighbour classifier for classification of speaker gender and height in figure 8. The classifier is fitted to time-averaged latent representations and Mel-features. We divide the height into three classes: below $175\,\mathrm{cm}$, above $185\,\mathrm{cm}$ and in-between. Compared to phonemes, the gender and height of a speaker are global attributes that affect the entire signal. In both cases, we see improved performance from using the learned latent space over Mel-features. Notably, $\boldsymbol{z}^2$ is outperformed by the Mel-features for gender identification which may indicate that $\boldsymbol{z}^2$ learns to ignore this attribute compared to $\boldsymbol{z}^1$.

We provide some additional latent space clustering of speaker gender in figure 9 and of speaker height in figure 10.

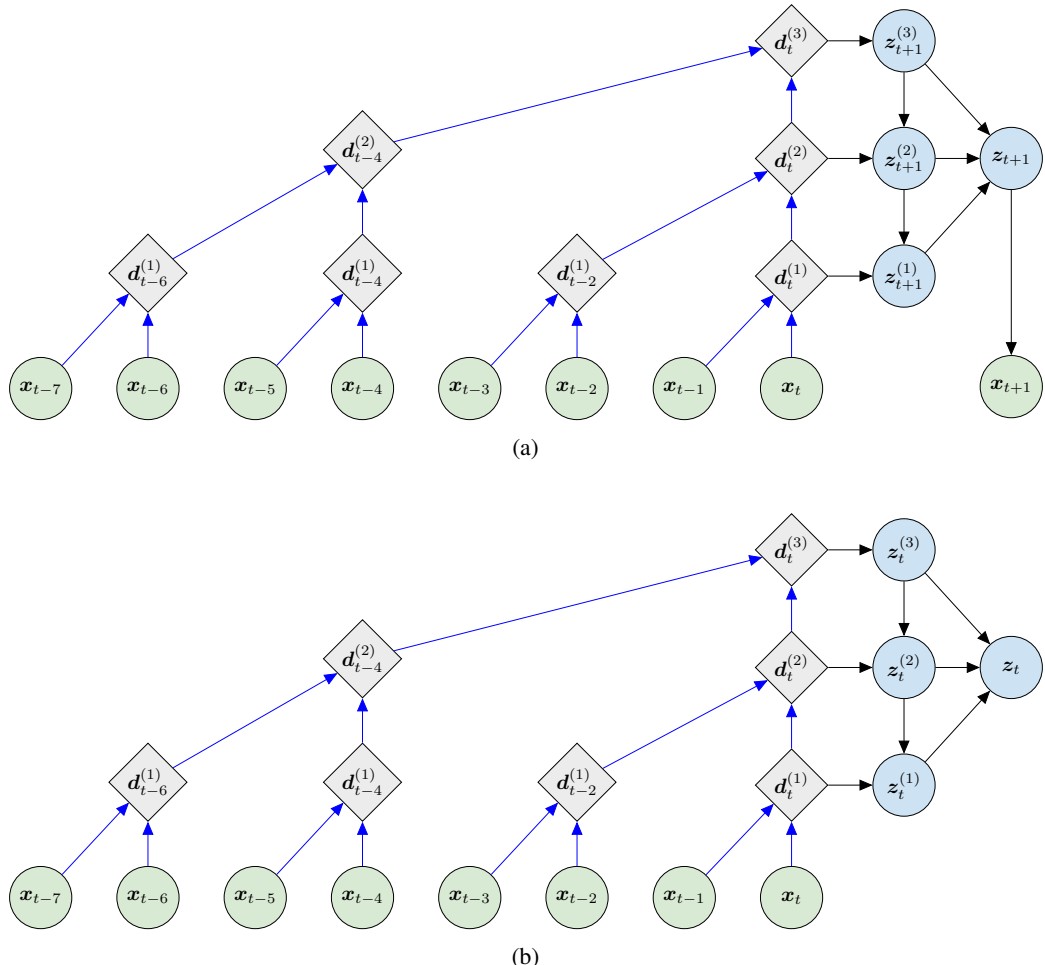

Figure 4: STCN (Aksan & Hilliges, 2019) generative model $p(\boldsymbol{x}, \boldsymbol{z})$ in (a) and inference model $q(\boldsymbol{z}|\boldsymbol{x})$ in (b) for a single time-step. The WaveNet autoregressive encoder is shared between generative and inference models. It is depicted here with only one stack of three layers in order to illustrate the dilated convolution with limited space. In practice, the model uses ten layers in each of five stacks/cycles resulting in a much larger receptive field. Importantly, the model parameterizes the five latent variables using the last deterministic representation $\boldsymbol{d}^{(l)}$ from each stack, i.e. only every fifth $l$ starting from $l = 5$ and ending at $l = 25$. Note that the generative model uses the prior to transform the WaveNet hidden states $\boldsymbol{d}_t^{(l)}$ into the latent variable $\boldsymbol{z}_{t+1}^{(l)}$ one step ahead in time compared to the approximate posterior which infers $\boldsymbol{z}_t^{(l)}$. Also note that $\boldsymbol{z}_t$ is constructed by concatenating all $\boldsymbol{z}_t^{(l)}$. The original paper explores setting $\boldsymbol{z}_t$ equal to $\boldsymbol{z}_t^{(1)}$. The best-performing STCN for speech, which also the one we implement, uses a WaveNet decoder to predict $\boldsymbol{x}_{t+1}$ from a sequence of $\boldsymbol{z}_t$ rather than a per-timestep transform. Blue arrows are shared between the inference and generative models.

All results presented here are obtained with a 2-layered CW-VAE trained on $\mu$-law encoded PCM similar to the one in table 1.

## L DISTRIBUTION OF PHONEME DURATION IN TIMIT

In figure 11 we plot a boxplots of the duration of each phoneme in the TIMIT dataset. We do this globally as well as for a single speaker to show that phoneme duration can vary between individual speakers.

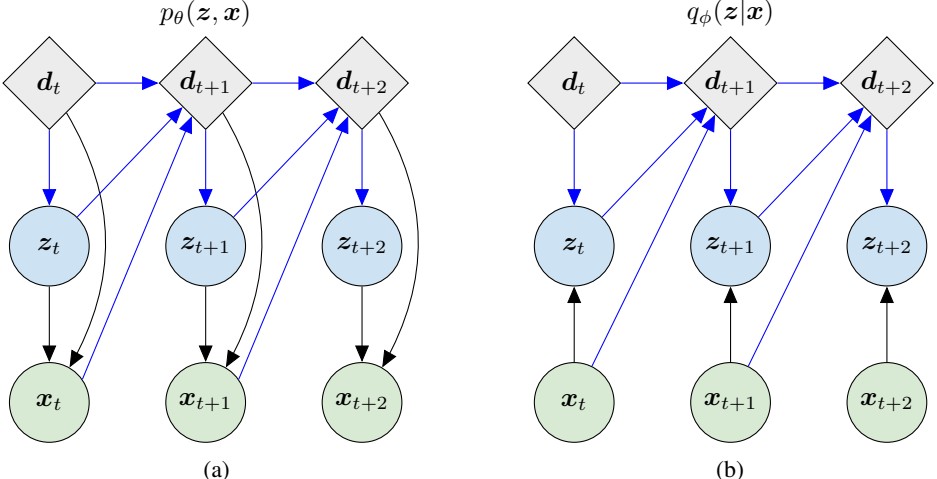

Figure 5: VRNN (Chung et al., 2015) generative model $p(\boldsymbol{x}, \boldsymbol{z})$ in (a) and inference model $q(\boldsymbol{z}|\boldsymbol{x})$ in (b) unrolled over three steps in the observed variable. Blue arrows are shared between the inference and generative models.

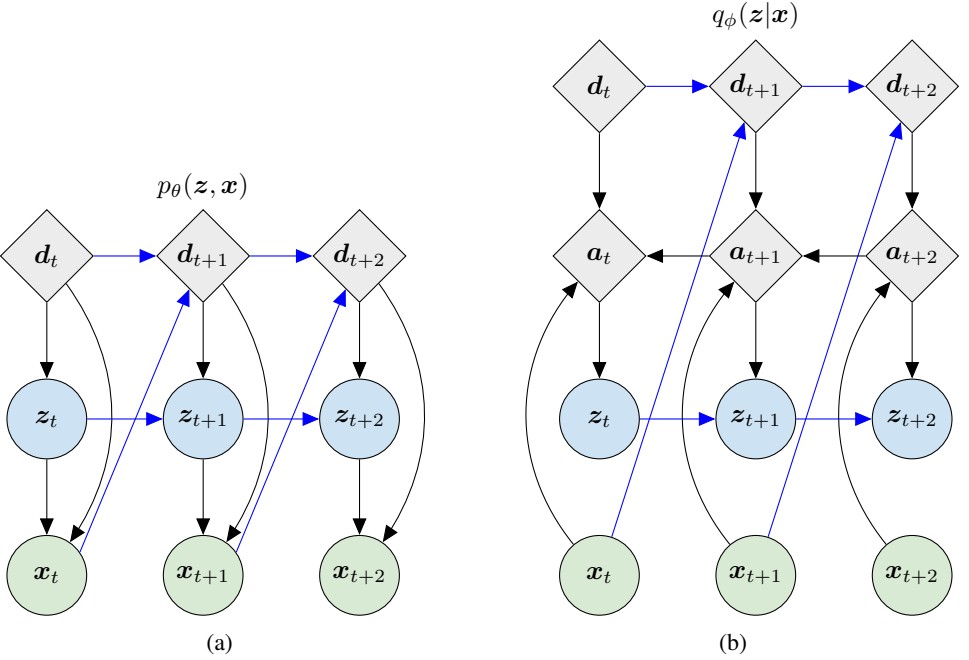

Figure 6: SRNN (Fraccaro et al., 2016) generative model $p(\boldsymbol{x}, \boldsymbol{z})$ in (a) and inference model $q(\boldsymbol{z}|\boldsymbol{x})$ in (b) unrolled over three steps in the observed variable. Blue arrows are shared between the inference and generative models.

A description of the phonemes used for the TIMIT dataset can be found at `https://catalog.ldc.upenn.edu/docs/LDC93S1/PHONCODE.TXT`.

# M MODEL SAMPLES AND RECONSTRUCTIONS

We provide samples and reconstructions for some of the models considered here at the following URL: `https://doi.org/10.5281/zenodo.5704512`. The samples are generated from the prior of Clockwork VAE, SRNN and VRNN and from a WaveNet by conditioning on pure zeros. All

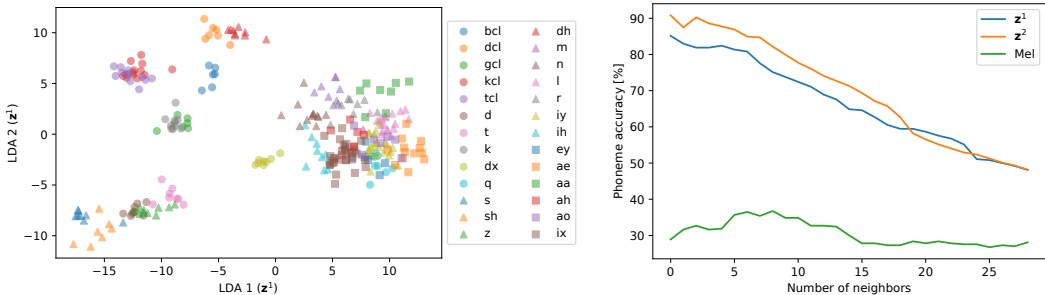

Figure 7: (left) Clustering of phonemes in a 2D Linear Discriminant Analysis (LDA) subspace of a CW-VAE latent space ($z^{(1)}$). (right) Leave-one-out phoneme classification accuracy for a KNN classifier at different $K$ in a 5D LDA subspace of a CW-VAE latent space.

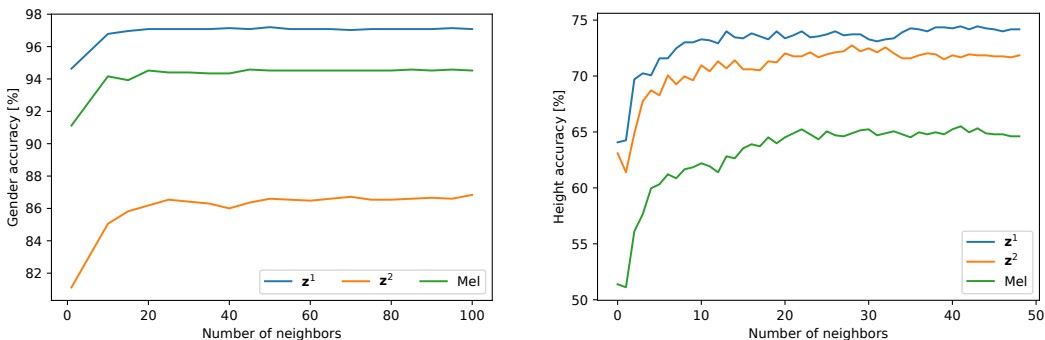

Figure 8: Leave-one-out $k$-nearest-neighbor accuracy with different $k$ for (a) the speaker's gender and (b) the height of male speakers (female speakers yield a similar result).

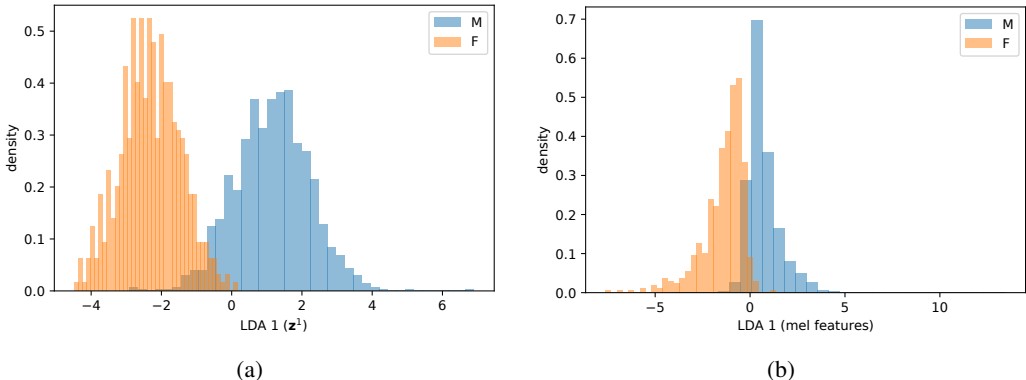

(a)                 (b)

Figure 9: Clustering of speaker gender in an one-dimensional linear subspace defined by a linear discriminant analysis of the CW-VAE latent space and of a time-averaged mel spectrogram. The total overlap is slightly smaller in the subspace of the CW-VAE latent space and the separation between the distribution peaks is larger.

models are configured as those reported in table 1. Importantly, the samples are unconditional. Hence they are *not* reconstructions inferred from a given input nor are they conditioned on any auxiliary data like text.

Although sample quality is a somewhat subjective matter, we find the quality of the unconditional Clockwork VAE to be better than those of our VRNN and SRNN. WaveNet is known to produce samples with intelligible speech when conditioned on e.g. text, but unconditional samples from WaveNet lack semantic content such as words as do VRNN, SRNN and Clockwork VAE.

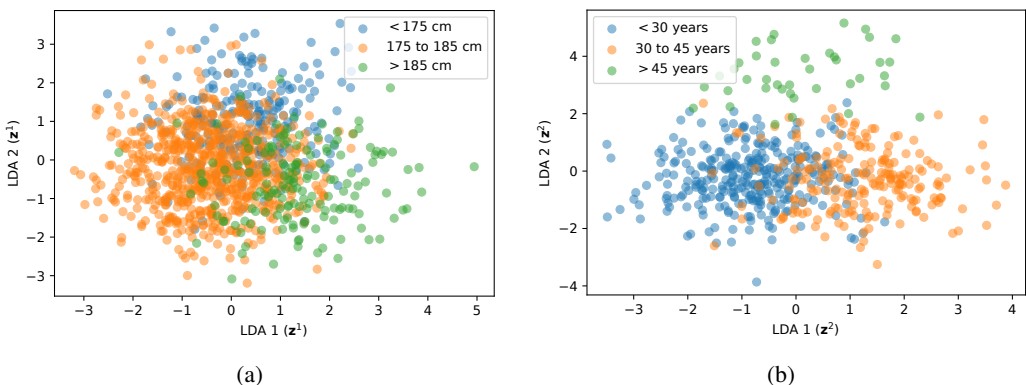

(a)                                                          (b)

Figure 10: (a) Clustering of speaker height for male speakers and (b) speaker age for female speakers in an two-dimensional linear subspace defined by a linear discriminant analysis of the CW-VAE latent space.

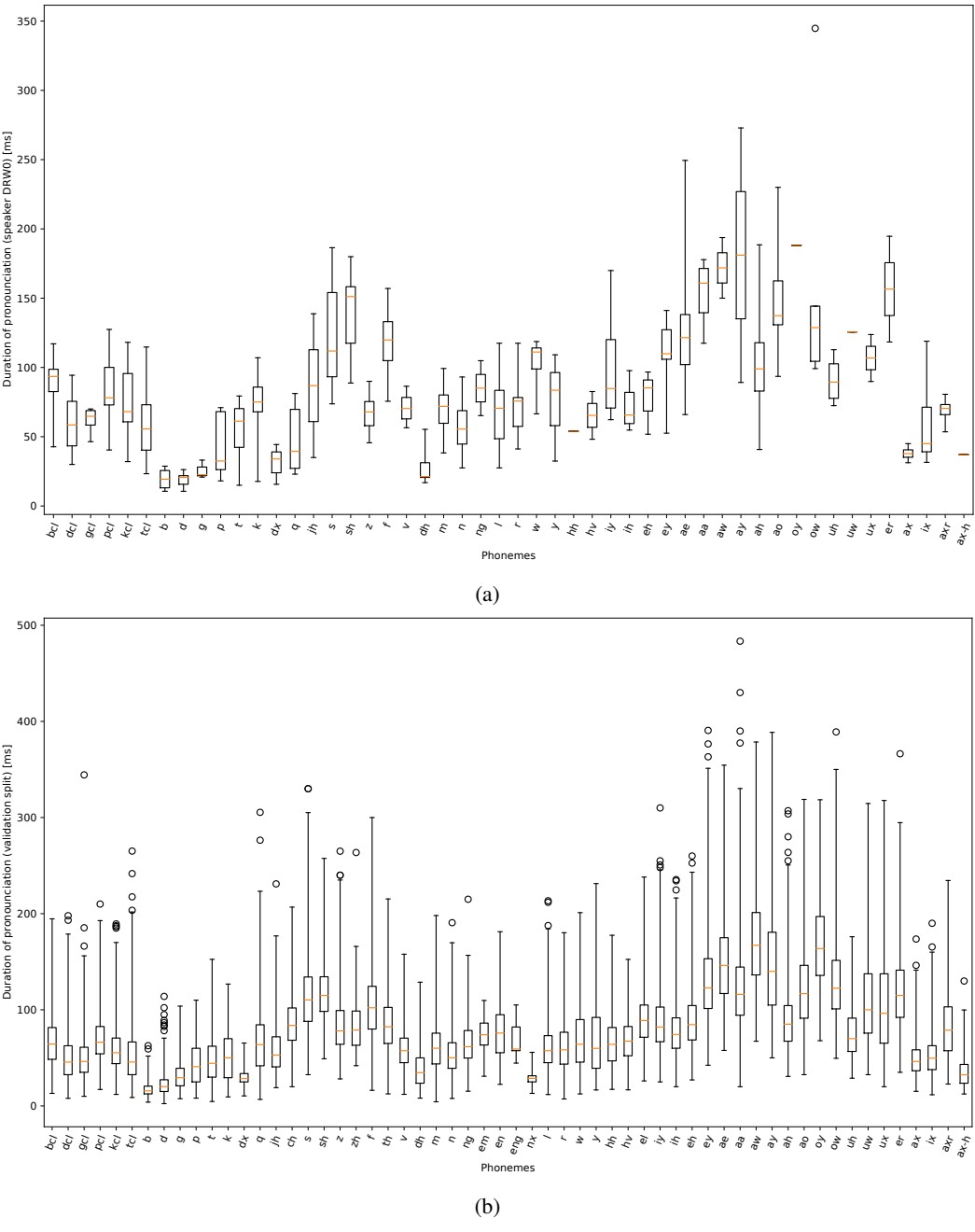

(a)

(b)

Figure 11: Boxplots of the duration of the pronunciation of phonemes in TIMIT for a specific speaker DRW0 in (a) and globally in (b). Not all phonemes are pronounced by speaker DRW0 over the course of their 10 test set sentences and hence they are missing from the x-axis compared to the global durations.

