# OpenReview forum: "Benchmarking Generative Latent Variable Models for Speech"
_ICLR.cc/2022/Workshop/DGM4HSD — ICLR 2022 DGM4HSD workshop Poster_

### Official Review · Reviewer_RX7R · 2022-03-17
**Good work with detailed exploration**

**Rating:** 7
**Confidence:** 3

**Review:**

The authors develop a benchmark for latent variable models on speech. They present not only the likelihood, but also the phoneme error rate as a more comprehensive measurement. Although it is hard to show all the ideas in limited pages, the authors still explain their work explicitly and add abundant appendix for reference.

There is one potential modification: In abstract the author suggest that Clockwork VAE outperform other models by using a hierarchy of latent variables, which is unclear to me in the main text as well as the chart. More explanation would be preferred.

Overall this is a good paper to appear on the workshop and the authors done solid work for benchmarking this field. So I would suggest to accept the paper.

---

### Official Review · Reviewer_WwHP · 2022-03-18
**Review of Benchmarking Generative Latent Variable Models for Speech**

**Rating:** 5
**Confidence:** 4

**Review:**

This paper benchmarks different VAE-based latent variable models for speech generation. The models benchmarked include VRNN (Chung et al., 2015), SRNN (Fraccaro et al., 2016) and STCN (Aksan & Hilliges, 2019). They further introduce a latent-variable model for speech based on the Clockwork VAE. The models are also evaluated on Phoneme recognition.

On low temporal resolution, i.e s=64, all models beat WaveNet. STCN performs the best on both modeling TIMNIT and Phoneme recognition that the authors attribute to the hierarchy of latent variables.

1. **Significance**: It is unclear to me what this benchmark offers in addition to already published results in terms of speech modeling. For example, Table 1 in (Aksan & Hilliges, 2019) already reports log-likelihoods on 4 different datasets with roughly the same ordering as reported by the authors. However, the results on downstream Phoneme recognition could be useful.
2. **Clarity**: The writing and description of the models considered in this work is mostly clear, especially alongside the diagrams in the appendix.  I would suggest the authors update the equations to use the deterministic representations rather than the latent variables, which might make it a bit more clearer. For eg, in Equation 2 $q(z | x) = \prod_{t=1}^T q(z_t | d_t, x_t)$ and $p(x, z) = \prod_{t=1}^T p(x_t | z_t, d_t) P(z_t | d_t)$
3. **Clarity**: One could also summarize the different design decisions in modeling the prior and posterior for the 4 different models in a Table. (for eg, GRU vs WaveNet, hierarchy vs non-hierarchy)
4. This is out-of-scope for this workshop submission but it seems that STCN has two components that are different from prior that could explain its superior performance (i.e the WaveNet encoder and latent-variable hierarchy), It would be nice to ablate them in future work.

My decision is borderline. The authors assess different models for speech generation on the TIMNIT dataset that has been done already in (Aksan & Hilliges, 2019). However, I think the results on the downstream phoneme recognition could be useful. If the authors opensource their code with all models readily available, that could also be useful to the community.

---

### Decision · Program_Chairs · 2022-03-25

Accept (Poster)